# A novel user clustering and efficient resource allocation in non-orthogonal mutliple access for IoT networks

Syed Muhammad Hamedoon[1]*, Jawwad Nasar Chattha[1], Muhammad Bilal[2]

**1** Department of Electrical Engineering, School of Electrical Engineering, University of Management and Technology, Lahore, Pakistan, **2** Center of Excellence in Intelligent Engineering Systems, Department of Electrical and Computer Engineering, King Abdulaziz University, Jeddah, Saudi Arabia

☯ These authors contributed equally to this work.
* f2017179002@umt.edu.pk

**Data Availability Statement:** All relevant data are within the manuscript.

**Funding:** This research work was funded by Institutional Fund Projects under grant no. (IFPIP:

## Abstract

Optimal resource allocation is crucial for 5G and beyond networks, especially when connecting numerous IoT devices. In this paper, user clustering and power allocation challenges in the downlink of a multi-carrier NOMA system are investigated, with sum rate as the optimization objective. The paper presents an iterative optimization process, starting with user clustering followed by power allocation of the users. Although the simultaneous transmission for multiple users achieves high system throughput in NOMA, it leads to more energy consumption, which is limited by the battery capacity of IoT devices. Enhancing energy efficiency by considering the QoS requirement is a primary challenge in NOMA-enabled IoT devices. Currently, fixed user clustering techniques are proposed without considering the diversity and heterogeneity of channels, leading to poor throughput performance. The proposed user clustering technique is based on the partial brute force search (P-BFS) method, which reduces complexity compared to the traditional exhaustive search method. After the user clustering, we performed optimal power allocation using the Lagrangian multiplier method with Karush-Kuhn-Tucker (KKT) optimal conditions for each user assigned to a subchannel in each cluster. Lastly, a deep neural network (DNN) based proposed P-BFS scheme is used to reduce resource allocation's complexity further. The simulation results show a significant improvement in the sum rate of the network.

## 1 Introduction

Mobile networks continuously evolve and adapt novel multiple-access standards to provide seamless and ubiquitous connectivity for a wide range of data-centric and bandwidth-hungry mobile services. Developing novel multiple-access techniques has improved spectrum and energy efficiency and allowed for high data rates. The fifth-generation (5G) networks already employ various novel techniques to enhance transmission speed and spectral efficiency [1].

One of the critical challenges in 5G is the shortage of radio resources, especially as more devices are anticipated to utilize network resources with the deployment of the massive

1822-135-1443). The authors gratefully acknowledge technical and financial support provided by the Ministry of Education and King Abdulaziz University, DSR, Jeddah, Saudi Arabia]. The funders had no role in study design, data collection and analysis, decision to publish, or preparation of the manuscript.

Internet of Things (IoT) [2]. Non-orthogonal multiple access (NOMA) has become a feasible and promising multiple access technique for 5G and beyond networks [3]. The fundamental principle of NOMA is to share a single resource block among multiple users. In a power-domain NOMA system for the downlink, the base station (BS) employs superposition coding (SC) to multiplex multiple signals intended for different users with varying transmission power levels on a non-orthogonal basis. NOMA ensures the quality of service and low latency transmission for many mobile devices [4, 5].

Resource allocation, which includes user grouping and power distribution, is an important factor in determining the performance of NOMA systems [6]. In NOMA, simultaneous transmission leads to an increased transmission rate. Different resource allocation techniques have been proposed to optimize power and user clustering in downlink Power Domain NOMA (PD-NOMA). Most of the research literature has measured the sum rate of NOMA using optimal power conditions and a user clustering approach. The complexity of traditional exhaustive BFS is high and takes more time as the input size grows compared to the proposed P-BFS. However, the proposed adaptive user clustering (AUC) approach in [7] is of significant importance as it searches through all possible combinations to find the best clusters, which can be computationally intensive. Similar related work in [8] relies on UC datasets obtained from the traditional Brute-Force search (BF-S) method, which is computationally intensive. The heuristic channel gain-based user clustering is proposed in [9, 10] which is simpler and faster than exhaustive search methods, it often sacrifices optimality, flexibility, and scalability, resulting in less efficient resource allocation and lower overall network performance. There is a significant gap in the literature regarding exploring a low-complexity user clustering approach for the energy efficiency perspective of a NOMA system that can achieve near-optimal performance.

This paper presents the optimal resource allocation solution for energy efficiency. We propose a novel user clustering and power allocation approach based on KKT conditions. Table 1 provides a list of the symbols used throughout this article. The major contributions of this work are listed below.

- A low-complexity user clustering approach is proposed to optimize the performance of IoT-based NOMA networks.

- A DNN-based proposed partial brute force search (P-BFS) resource allocation scheme is presented further to reduce the complexity of the proposed P-BFS method.

- Optimal power allocation for the IoT devices in the respective clusters is calculated according to the QoS requirements. This power allocation is based on KKT conditions to achieve high energy efficiency in downlink PD-NOMA.

- The results of the proposed and DNN-based P-BFS resource allocation schemes are compared with those of fixed and optimal power allocations. The findings clearly show an improved system sum rate using optimal power allocation. The performance of the proposed P-BFS-based user clustering techniques is very close to that of the exhaustive BFS and superior to the other channel gain-based user clustering technique.

## 1.1 Related work

Researchers have recently explored numerous research works on user clustering techniques and power allocation for traditional NOMA networks. Machine learning approaches for resource allocations are also being proposed for networks beyond 5G. The author, in [7], proposed artificial neural network (ANN) and deep neural network (DNN) based used

**Table 1. List of notations and mathematical symbols.**

| Notation | Description |
|---|---|
| n | Number of IoT devices |
| j | Cluster number from set { 1, 2, . . ., c} |
| i | Set of user from {1,2,. . .,n} |
| B | Total system bandwidth |
| $P_i^j$ | Transmit power of the $i^{th}$ user in cluster $j$ |
| $\omega_j$ | Resource block of the $j^{th}$ cluster |
| $U_{\omega_j}$ | User share resource block in cluster $j$ |
| $P_{\max}$ | Maximum transmission power of the base station |
| $h_i^j$ | Channel gain of the $i^{th}$ user inside $j$ cluster |
| $\sigma_n^2$ | Variance of AWGN |
| $P_c$ | Circuit Power |
| $R_i^j$ | Rate of $i^{th}$ user in $j^{th}$ cluster |
| $R_{\min}$ | Threshold to guarantee minimum data rate transmission requirement |
| $\epsilon$ | The minimum power gap for successful SIC |
| $B_K$ | Bell number to partition K number of users |
| $V_k^j$ | Partial partition $k$ for cluster $j$ |
| $R_k^j$ | Rate for each partial partition for cluster $j$ |
| $p_b$ | Power is allocated to each resource block |
| b | Bandwidth of each resource block |
| L | Lagrange function |
| $\lambda_i, \mu_i, \pi_i$ | Lagrange variables |
| Z | Iteration index |
| $\gamma_i^j$ | Signal-to-interference-plus-noise ratio (SINR) of $i$- th user from cluster $j$ |
| $n^j$ | Number of user belongs to cluster $j$ |
| I | Number of input layer nodes |
| H | Number of hidden layer nodes |
| V | Number of output layer nodes |
| x | Input vector for DNN based P-BFS |
| Z | Output vector for DNN based P-BFS |
| p | Hidden layer index |
| N | Number of hidden layer nodes |
| $W^p$ | Weight matrix that links $(p-1)^{th}$ to $p^{th}$ layer |
| $f^p$ | Activation function of the p-th layer |
| $b^q$ | Bias vector for the $p^{th}$ layer |
| $a^p$ | Output vector of hidden layer |
| $Z_k^*$ | Predicted sum rate at the output of DNN based P-BFS |
| $H_\alpha$ | Cost function |

clustering in downlink NOMA. Using the B-FS method, an adaptive user clustering approach is proposed by finding all the clustering combinations and selecting the best cluster to achieve maximum throughput. The proposed technique significantly reduces the complexity of the resource allocation algorithm as compared to the other optimal approaches. The authors in [8] present the deep neural network (DNN) based user clustering technique to enhance the network's performance. The optimization of hyperparameters improves the learning ability of the neural network. The author in [9] proposed a user clustering technique based on higher channel gain difference. The optimal power allocation of each cluster is

calculated using the Lagrangian multiplier method with Karush–Kuhn–Tucker (KKT) conditions. The numerical results show that the proposed clustering and power allocation schemes significantly improve the throughput of the downlink PD-NOMA. The user clustering technique proposed in [10] is based on higher channel gain difference. After the user clustering, the power allocation is performed using DNN and DL for each subcarrier to improve the sum rate. The results show the comparison between the fractional and fixed transmit power using a deep learning approach.

In [11], the author proposed an iterative algorithm to optimize power allocation and user clustering for IoT networks. A resource optimization scheme is proposed for narrow-band NOMA-enabled IoT in [12]. The joint solution is obtained using a heuristic algorithm to allocate resources and maximize the throughput. The authors in [13] present the optimization technique to reduce the maximum access delay in NOMA-enabled IoT networks. Joint user clustering and power allocation were adopted for the IoT devices to fulfill the QoS requirements. The graph-based method was used to address the user scheduling problem, and the iterative algorithm was adopted to address the power control problem.

In [14] again, the authors present a novel resource management for NOMA-based IoT. The scheme is a suboptimal NOMA power allocation scheme with low complexity, employing Karush-Kuhn-Tucker (KKT) conditions, and compared with the conventional orthogonal multiple access (OMA) included for comparison. The proposed scheme restricted the number of frequency blocks, offering an optimal method for allocating power and frequency blocks in an IoT network. A user clustering approach is proposed in [15], based on SARSA Q-learning and Deep reinforcement Learning to allocate resources in a multicell uplink IoT NOMA system. The method of optimal power allocation is proposed to improve the sum rate by optimizing the constraints of quality of service (QoS) [16]. The optimal joint subcarrier and power allocation method is discussed in [17], based on the MISO-NOMA network to optimize the weighted sum throughput by considering the QoS requirements.

In [18], a user pairing scheme based on Stackelberg game theory is proposed to maximize the sum rate and ensure QoS requirements. First, the solution in the case of two users, where the possibility of dividing the transmit power between the two users is shown. An iterative approach is then described based on the developed algorithm for a related two-user scenario. In [19], the author proposed an optimal user clustering technique based on the Hungarian algorithm for 2-user pairing. The closed-form solution for the optimal power allocation is obtained using KKT conditions to improve the sum rate of PD-NOMA downlink. The coordinated multipoint (CoMP) transmission is used in green non-orthogonal multiple access (NOMA) based IoT to improve the network capacity that enables access points (APs) to serve one device simultaneously without cutting off the association with another device. This technique effectively saves the transmission power and meets the QoS requirements of all the devices [20]. An iterative technique is proposed in [21] to optimize power allocation and energy efficiency in mmWave massive MIMO-NOMA systems. The precoding scheme employed is a hybrid one which however takes into consideration the number of RF chains available at the base station. Based on this, a novel iterative power allocation has been developed to achieve an optimum power in order to enhance the energy efficiency. The author in [22] presented a hybrid precoding-based user clustering approach in a single-cell downlink system for mmWave-MIMO-NOMA. The proposed technique optimized the power allocation using power splitting factors to maximize the sum rate. The author in [23] addresses the resource allocation problem in the number of clusters by finding the optimal power solution to maximize the spectral efficiency (SE) of the downlink single-cell Hybrid-NOMA system. The power budget is fully optimized in clusters to maintain the minimum power requirement to satisfy the QoS demand of users inside clusters.

The performance of NOMA in mmWave communication is evaluated in terms of spectral efficiency (SE) and energy efficiency (EE) discussed in [24]. Firstly, a user grouping approach is provided based on the channel correlations of a user. Then a new hybrid beamforming and power allocation problem is introduced to achieve the maximum sum rate while satisfying the minimum rate requirement for each user. The proposed solution for the mmWave-NOMA for user grouping, joint power allocation, and hybrid beamforming exploits all the benefits of achieving a higher sum rate and energy efficiency than the benchmark scheme but with a higher computational complexity. In [25], the authors consider a NOMA heterogeneous network (HetNet) in mmWave communications, comprising of a macro-cell and small cell tiers connected via wireless backhaul. A user grouping algorithm is proposed to significantly simplify the clustering process, with each cluster consisting of highly correlated users to suppress inter-cluster interference. In [26], the author proposed a fuzzy c-means(FCM) algorithm based on QoS parameters for user clustering. The results show a significant improvement in the sum rate and energy efficiency of the mMIMO-NOMA-based network. A joint user clustering, ordering, and beamforming methods are proposed in [27] that reduces the number of clusters in mmWave-NOMA ABF system that serve a set of users based on their successive interference cancellation (SIC) decoding capability constraints.

A novel method called NOMA-PKKT-HNG is presented in [28] to address power allocation and user pairing challenges in downlink NOMA systems, aiming to maximize the system's total rate. The non-convex constrained problem is solved by constructing optimal power allocation's closed-form solutions using Karush-Kuhn-Tucker (PKKT) conditions. In [29], authors consider underwater communication challenges which are bandwidth limited, with low data rate, large propagation delay, and high bit error rate (BER). The coding scheme improved the rate of error-corrected data while effectively reducing the BER in the communication system. A recent work [30] introduced a dynamic user clustering (DUC) scheme. This scheme involves selecting an appropriate number of clusters and organizing NOMA users into these clusters. Consequently, the cluster size is dynamically adapted to performance requirements.

## 2 System model and problem formulation

We consider a PD-NOMA downlink in which the transmission power of the base station (BS) is divided among the $n$ IoT users, as shown in Fig 1. The $n$ number of IoT users is randomly distributed inside a cell with a radius $d$, which a single BS serves. The $n$ number of users are divided into $j$ clusters, where $j \in \{1, 2, \ldots, c\}$. Within these clusters, there are $i$ users where, $i \in \{1, 2, 3 \ldots, n\}$. It is assumed that the BS has complete information about the CSI for its associated users. The channels between the BS and IoT users are modeled using Rayleigh fading as in [14].

The maximum downlink transmission power of the BS is represented as $P_{\max}$ and the power allocated to the $i^{th}$ user in $j$ cluster is denoted as $P_i$, where $i \in \{1, 2, 3 \ldots, n\}$.

$$P_1^j + P_2^j +, \ldots, P_n^j \leq P_{\max} \tag{1}$$

The channel gain of $i^{th}$ user inside the $j$ cluster is represented by $h_i^j$ in Eq 2

$$\left| h_1^j \right|^2 > \left| h_2^j \right|^2 >, \ldots, > \left| h_n^j \right|^2, \forall i \in \{1, 2, \ldots, n\} \tag{2}$$

Using NOMA with SIC, BS can transmit individual messages to multiple users within a cluster in the same subchannel. The BS broadcasts the superposition of individual messages to all the IoT users inside a particular cluster. The basic principle of SIC is to reduce interference

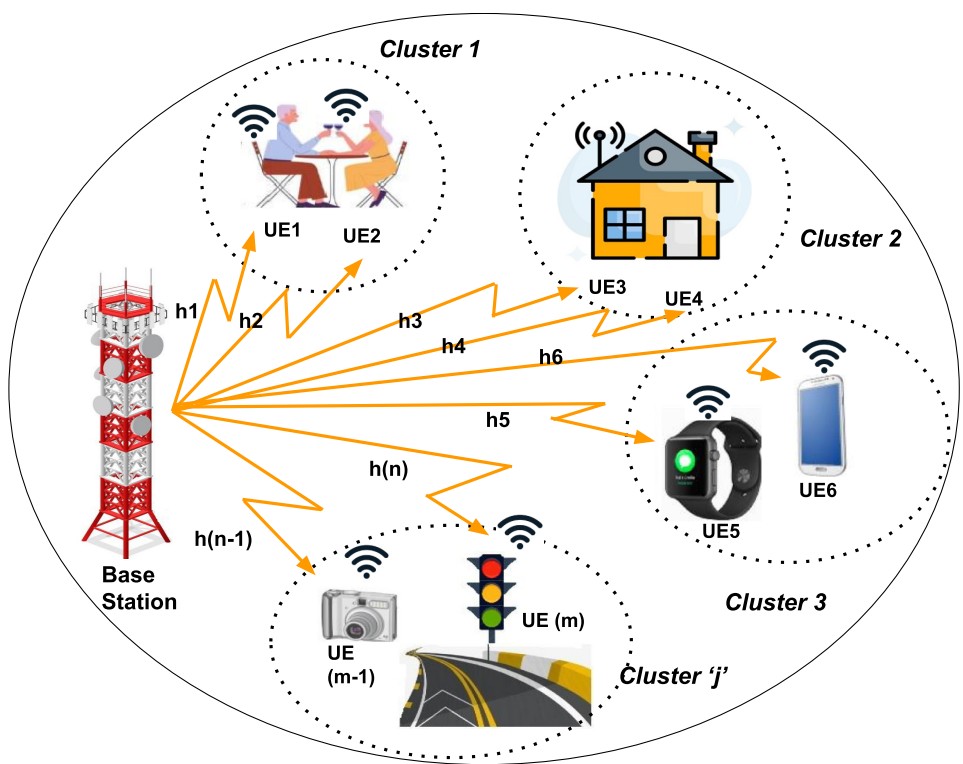

**Fig 1. System model for NOMA enabled IoT.**

among IoT users by iteratively decoding them. The IoT users' gain is sorted in descending order on each subchannel as mentioned in Eq 2. With SIC, each user recovers their signal by successively reducing interference from other low-channel gain users in the same clusters. First, it decodes the messages of 1$^{st}$ users while considering the superposition transmission of the remaining users as interference. Next, the decoded transmission of the 1$^{st}$ user is removed from the superimposed received signal. The remaining user messages are iteratively decoded. This iterative process of SIC decoding continues until all the users get their desired signal.

We assume a total system bandwidth $B$ and the maximum transmission power of the BS is $P_{\max}$ which is equally divided into $k$ frequency resource blocks, and their bandwidth is $b = \frac{B}{k}$ and the power is allocated to each resource block is $p_b = \frac{P_{max}}{k}$. The resource blocks is denoted by $\omega_j$ which is allocated to the $j^{th}$ cluster and $\omega_1 + \omega_2 +, \ldots + \omega_c = k$. The users inside the cluster should share the same subchannel $\omega_j$. We consider the $n$ user must be grouped in to $j$ cluster. Each cluster can share the same resource block for the $n$ number of IoT devices using non-orthogonal scheduling, while other clusters can work independently.

With SIC decoding at the receiver, the signal-to-interference plus noise ratio (SINR) for each IoT device inside the cluster is represented by $\gamma_i^j$ as in Eq 3.

$$\gamma_i^j = \frac{P_i^j \left| h_i^j \right|^2}{\omega_j + \sum_{k=1}^{i-1} \left| h_i^j \right|^2 P_k^i + \sigma_n^2}, \forall i \in n \tag{3}$$

where, $P_i^j$ represents the BS's transmission power for the $i^{th}$ IoT user belonging to the cluster $j$. Similarly, the channel gain of the $i^{th}$ IoT user in the $j^{th}$ cluster is represented by $\left| h_i^j \right|^2$. Also,

$\sigma_n^2$ represents the additive white Gaussian noise(AWGN) with zero mean and unit variance. The achievable throughput for the $i^{th}$ IoT devices assigned to the $j^{th}$ cluster can be expressed as

$$R_i^j = \omega_j b \log_2(1 + \gamma_i^j) \tag{4}$$

where $R_i^j$ is the achievable rate of each IoT device which is measured in bits per second (bps). Energy efficiency is the ratio of throughput and total power consumption. The IoT users' achievable rate and power allocation can be used to define the maximum energy efficiency $EE_{\max}$ in the downlink NOMA system as in Eq 5.

$$EE_{\max} = \frac{R_i^j}{P_c + \sum_{i=1}^{n^j} P_i^j} \tag{5}$$

where $P_i^j$ is transmission power of the $i^{th}$ user in the $j^{th}$ cluster and $P_c$ is the circuit power which is assumed as 1watt. Where $n^j$ represents the $n$ number of users for the particular $j$-th cluster. The maximum sum rate and energy efficiency for $n$ IoT devices in the NOMA system must also follow the following constraints.

- The maximum power transmission constraint of the BS is

$$\sum_{i=1}^{n^j} P_i^j \leq P_{\max} \tag{6}$$

- The maximum power transmission constraint of each cluster is

$$\sum_{i=1}^{n^j} P_i^j \leq \omega_j P_b, \forall j \in \{1, \ldots, c\} \tag{7}$$

- The necessary power constraints for each IoT user to perform SIC in cluster $j$ is

$$\sum_{j=1}^{c} \left( \sum_{k=1}^{i-1} P_k^j + \frac{\epsilon}{\left| h_{i-1}^j \right|^2} \leq P_i^j \right), \quad \forall_i \in \{2, \ldots, n^j\} \tag{8}$$

- The minimum data rate transmission requirement of each IoT device is

$$\sum_{i=1}^{n^j} \omega_j b \log_2(1 + \gamma_i^j) \geq R_{\min} \tag{9}$$

$$\forall_i \in \{1, 2, \ldots, n^j\}$$

## 3 Proposed resource allocation techniques

This section presents the proposed practical resource allocation problem, which divides the IoT users into multiple clusters and employs NOMA within each cluster. The first step

involves proposing a user clustering scheme based on the P-BFS method. The second step performs optimal power allocation by developing an optimization problem and employing KKT conditions. The third step reduces the complexity of the proposed P-BFS scheme by using DNN-based clustering.

## 3.1 Proposed partial brute force search (P-BFS) method for user clustering

A user clustering algorithm based on a P-BFS search method is proposed. Users are paired based on the channel gain difference, with users closer to the base station having better channel gain than those farther away. The significant advantage of the P-BFS method is that it reduces the complexity of the typical exhaustive search method, i.e., the brute-force search (BFS). In the BFS method, there would be $(n)!$ possible combinations and the total cost for calculating the path would be $O(n)$. The total time complexity would be $O(n!)$. The possible combinations of clustering outcomes $B_K$ are determined by using the Bell formula [31], which is mentioned in Eq 10.

$$B_K = \sum_{k=0}^{K-1} \binom{K-1}{k} B_k \qquad (10)$$

Here, $B_K$ represents the Bell number for partitioning $K$ number of users, where $B_0 = B_1 = 1$, $B_2 = 2$, $B_3 = 5$, $B_4 = 15$, and so on. The set of $n$ IoT users sharing the resource block $\omega_j$ in cluster $j$ can be represented as $U_{\omega_j} = \{U_1, \omega_j, U_2, \omega_j, \ldots, U_n, \omega_j\}$ where $n \leq K$. Since a cluster $j$ consists of a set of users sharing the same resource blocks. We extract the partial partitions $V_k$ which are based on 2 user channel gain difference $|h_1|^2 > |h_2|^2$ using Algorithm 1.

The possible combinations for $n = 10$ are 115975 using Eq 10, including the disjoint set. The set of 2 user pair possible combinations is calculated by using Eq 11

$$\text{Possible } k \text{ number of user pairs} = \frac{(2k)!}{2^k.k!} \qquad (11)$$

where $n = 2k$, the number of ways to form $k$ pairs. When $n = 10$ the total possible combination for 2 user pair is 945 this reduces the overall combinations as compare to Eq 10. The optimal 2-user grouping is obtained for the number of partial partitions in each cluster using Algorithm 1, shown in Fig 2. When $n > 10$, the possible combinations computed through the Exhaustive B-FS method using Eqs 10 and 11 gradually increase their complexity. The time complexity of traditional BFS scheme is $O(n!)$ which exponentially grows for the large number of $n$. The time complexity of the proposed user clustering based on P-BFS given in Algorithm 1 is $O(n.2^n)$. Therefore, the time complexity of traditional BFS will be higher and take more time as the input size grows compared to a proposed P-BFS Algorithm 1. The detailed steps of the proposed P-BFS method for user clustering are mentioned in Algorithm 1.

We consider the example of making clusters in groups of 2 users for selected numbers of users $n = 6$ and $n = 18$ by using the proposed P-BFS-based method for user clustering. Table 2 represents the number of partial partitions created for $n = 6$. We measured the sum rate for 2 user groups in each partial partition. We found from Table 2 that the sum rate is maximum for the partial partition number 10. We select the partition whose sum rate is relatively higher than the others. So we find optimal user pairing for $n = 6$ obtained from the Table 2 is $\{U_1, U_4\}$, $\{U_2, U_5\}$, $\{U_3, U_6\}$. Where $\psi$ in Algorithm 1 represents the maximum number of users to accommodate inside the cluster $j$. We assume $\psi = 10$ for user clustering in Algorithm 1. We consider another example to make clusters when $n = 18$ using Algorithm 1, the total number of clusters created is 2, and the number of partial partitions for 2 user grouping is 945 and 105 for each cluster. The possible number of user pairs for $n = 18$ using traditional BFS method

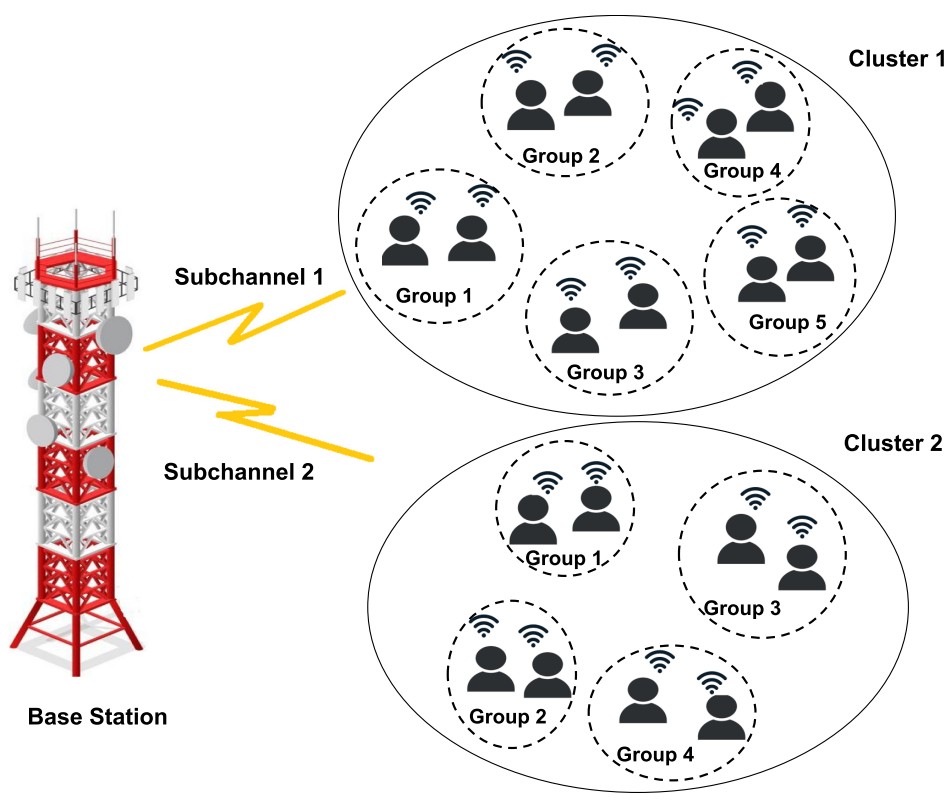

**Fig 2. Partial exhaustive search method for *n* = 18.**

from Eq 11 is 34, 650, which is significantly larger in numbers as compared to proposed P-BFS Algorithm 1 which creates 2 clusters and the number of partial partition ($V_k$) in each cluster is 945 and 105. The proposed P-BFS Algorithm 1 selects the partition for each cluster, which returns a higher sum rate and neglects the other partial partitions ($V_k$), as shown in Fig 2.

**Table 2. Number of partial partitions created for *n* = 6.**

| Partial Partition Number | Possible User Grouping in each Partial-Partition | Sum Rate (Mbps) |
|---|---|---|
| 1 | {1,2}{3,4}{5,6} | 31.598573 |
| 2 | {1,3}{2,4}{5,6} | 32.187079 |
| 3 | {1,4}{2,3}{5,6} | 31.872688 |
| 4 | {1,2}{3,5}{4,6} | 32.101850 |
| 5 | {1,2}{3,6}{4,5} | 31.533868 |
| 6 | {1,3}{2,5}{4,6} | 31.074528 |
| 7 | {1,3}{2,6}{4,5} | 31.051044 |
| 8 | {1,5}{2,3}{4,6} | 30.168268 |
| 9 | {1,6}{2,3}{4,5} | 31.433566 |
| **10** | **{1,4}{2,5}{3,6}** | **32.237459** |
| 11 | {1,4}{2,6}{3,5} | 31.557039 |
| 12 | {1,5}{2,4}{3,6} | 31.879133 |
| 13 | {1,6}{{2,4}{3,5} | 31.620771 |
| 14 | {1,5}{2,6}{3,4} | 31.034256 |
| 15 | {1,6}{{2,5}{3,4} | 31.442131 |

## 3.2 Optimal power allocation for proposed user cluster using KKT

Our objective function is to improve the sum rate of different numbers of IoT users according to the QoS requirements. To obtain the optimal power allocation solution for sum rate maximization, the resource management problem for achieving the desired objective is represented in Eq 12.

$$\max_{P_i^j} \sum_{i=1}^{n^j} \omega_j b \log_2(1 + \gamma_i^j) \tag{12}$$

We formulate the problem by considering the following constraints using Karush-Kuhn-Tucker optimal conditions. [32] as shown in Eq 13.

$$
\begin{cases}
C1 : \sum_{i=1}^{n^j} P_i^j \leq P_{\max}, \\[2ex]
C2 : \sum_{j=1}^{c} \left( \sum_{k=1}^{i-1} P_k^j + \dfrac{\epsilon}{\left| h_{i-1}^j \right|^2} \leq P_i^j \right) \\[2ex]
\qquad \forall_i \in \{2, 3, \ldots, n^j\} \\[2ex]
C3 : \sum_{i=1}^{n^j} \omega_j b \log_2(1 + \gamma_i^j) \geq R_{\min} \\[2ex]
\qquad \forall_i \in \{1, 2, \ldots, n^j\}
\end{cases}
\tag{13}
$$

The constraint $C1$ represents the maximum power transmission of the cluster represented in Eq 13. The necessary power constraint for the efficient SIC process of each IoT device is represented by $C2$ in Eq 13. The constraint $C3$ represents the minimum data rate requirement of each IoT user in Eq 13.

The optimization problem can be solved by the Lagrangian multiplier method [30, 33]. The optimal problem can be expressed as

$$
\begin{aligned}
L(P_i^j, \lambda, \mu, \pi) = & \sum_{i=1}^{n^j} R_i^j + \sum_{i=1}^{n^j} \lambda_i (R_{\min} - R_i^j) \\[2ex]
& + \sum_{i=1}^{n^j} \mu_i \left( \sum_{i=1}^{n^j} P_i^j - P_{\max} \right) \\[2ex]
& + \sum_{i=2}^{n^j} \pi_i \left( \sum_{k=1}^{i-1} P_k^j + \frac{\epsilon}{\left| h_{i-1}^j \right|^2} - P_i^j \right)
\end{aligned}
\tag{14}
$$

Where $\lambda, \mu, \pi$ are the Lagrangian multipliers and after the derivation of the Lagrangian

multiplier, by applying KKT, we compute the partial derivative of Eq 14 as follows:

$$\frac{\partial L}{\partial P_i^j} = \frac{\partial}{\partial P_i^j} \left[ \sum_{i=1}^{n^j} R_i^j + \sum_{i=1}^{n^j} \lambda_i \left( R_{\min} - R_i^j \right) \right.$$

$$+ \sum_{i=1}^{n^j} \mu_i \left( \sum_{i=1}^{n^j} P_i^j - P_{\max} \right) \tag{15}$$

$$\left. + \sum_{i=2}^{n^j} \pi_i \left( \sum_{k=1}^{i-1} P_k^j + \frac{\epsilon}{\left| h_{i-1}^j \right|^2} - P_i^j \right) \right]$$

**Algorithm 1** Proposed P-BFS Based User Clustering

```
Input: the n number of IoT devices which is served by the base station
in each cluster j.∀i ∈ {1, 2, …, n} and ∀j ∈ {1, 2, …, c}
Output: Cost matrix = (V_k, R_k)^j
1: Create Clusters j
2:   If (0 > n/ψ < 1)
3:   Cluster j ← 1
4:   If (1 > n/ψ < 2)
5:   Cluster j ← 2
6:   Until If (c − 1 > n/ψ < c)
7:   Cluster j ← c
8: Find possible number of partitions B_K for cluster j using Eqs 10 and
   11
9: Find k partial partitions V_k for 2-user grouping on the basis of
   channel gain difference for j cluster computed from Step 2 to Step
   8 which is expressed as V_k^j = {Group_1, Group_2, …, Group_n}^j
10: For j = 0 to c do
11:   For k = 0 to n do
12:   Find Rates R_i of each 2 pair of user in V_k partial partitions by
      using Eq 4 for each cluster j
13:   end for
14: end for
15: Select those V_k partitions for cluster j which have maximum
    throughput and remove the other partitions.
16: Obtain optimal user clustering cost matrix (V_k, R_k)^j for their max-
    imum sum rate in V_k partial partitions of each cluster j.
17: End
```

After some derivations and modifications of Eq 15 by setting $\frac{\partial L}{\partial P_i^j} = 0$, we get

$$\frac{\lambda_i \Gamma_i^j}{\ln 2(\omega_j + P_i^j \Gamma_i^j)} - \gamma_k^j + \mu_i - \pi_i = 0 \tag{16}$$

The values of $\Gamma_i^j$ and $\gamma_k^j$ are

$$\Gamma_i^j = \left( \frac{\left| h_i^j \right|^2}{\omega_j + \left| h_i^j \right|^2 \sum_{k=1}^{i-1} P_k^j + \sigma_n^2} \right) \tag{17}$$

$$\gamma_k^j = \sum_{k=1}^{i-1} (1 + \lambda_k) \left( \frac{\Gamma_k^j \left| h_k^j \right|^2}{\ln 2 \left( 1 + \gamma_k^j \right) \sum_{l=1}^{k-1} P_l^j |h_k^j|^2} \right) \tag{18}$$

Now for solving for $P_i^j$, the optimal power equation can be written as

$$\left(P_i^j\right)^* = \left[\frac{(1+\lambda_i)}{\gamma_k^j + \pi_i - \mu_i} - \frac{1}{\Gamma_i^j}\right]^+ \tag{19}$$

We iteratively update the value of $\lambda_i$, $\mu_i$, $\pi_i$ by using sub-gradient method as [34, 35].

$$\lambda_i(z+1) = \left[\lambda_i(z) + \psi(z)\left(R_{\min} - \sum_{i=1}^{n^j} R_i^j\right)\right]^+, \forall j \tag{20}$$

$$\mu_i(z+1) == \left[\mu_i(z) + \psi(z)\left(\sum_{i=1}^{n^j} (P_i^j - P_{\max})\right)\right]^+, \forall j \tag{21}$$

$$\pi_i(z+1) = \left[\pi_i(z) + \psi(z)\left(\sum_{k=1}^{i-1} P_k^j + \frac{\epsilon}{\left|h_{i-1}^j\right|^2} - P_i^j\right)\right]^+, \forall j \tag{22}$$

In each iteration of $z$, we first update $\lambda_i$, $\mu_i$ and $\pi_i$ by using $P_i^*$ and then use optimal values of $\lambda_i$, $\mu_i$ and $\pi_i$ to obtain optimal value of power allocation $P_i^*$. The complete steps of KKT-based power allocation is mention in Algorithm 2.

### 3.3 DNN based proposed P-BFS user clustering

A user clustering (UC) dataset is obtained based on the proposed P-BFS Algorithm 1. The next step involves applying a DNN to the UC dataset to further reduce the complexity of the proposed P-BFS algorithm. To train the DNN model, the dataset consists of transmit powers, channel gains, and their corresponding sum rate values, which are generated using our proposed Algorithm 1. In the $K$ user NOMA system, the total $3K$ attributes in which $K$ number of channel gains, transmit powers, and throughput. Although the computational complexity of the exhaustive BFS is excessively high, rendering it impractical for real-world implementation due to selecting the upper limit for the NOMA system. In proposed P-BFS Algorithm 1, once the number of partial partitions is created for each cluster $j$, the goal is to find the optimal partition for each cluster in terms of the maximum sum rate. The proposed P-BFS-UC data is fed to the DNN and to predict the throughput for each partial partition in cluster $j$. The basic benefit of DNN-based user clustering is that it saves time and increases the efficiency of the system.

**Algorithm 2** KKT-based power allocation

```
Input: Initialize n, j, Pᵢʲ, Pₘₐₓ, ωⱼ and z = 0
Output: f(j, P)
1: for j = 1 to c do
2:    for i = 0 to n
3:    Compute (Pᵢʲ)* using Eq 19
4:    Compute the Lagrangian multipliers λᵢ, πᵢ and μᵢ using Eqs 20, 21
      and 22 using sub-gradient method.
5:       Select the step size as ψ(z) = z/(2z+1)
6:       Calculate the power Pᵢʲ to update the Lagrange multipliers λᵢ, πᵢ
         and μᵢ
7:       iterate z = z + 1
8:    if Pʲ satisfies all the optimal KKT conditions then
9:        q ← Σᵢ₌₁ⁿʲ ωⱼb log₂(1 + γᵢʲ)
```

```
10:        f(j, P) ← max (f(j, P), q)
11:        end if
12:     end for
13:   end for
```

**3.3.1 Working principle of DNN-based scheme.** Fig 3 shows the basic architecture of DNN. It consists of three layers, namely an input layer, a hidden layer, and an output layer. It is a completely connected feed-forward neural network in which every node is interconnected with all nodes in the adjacent layer, and there are no connections between nodes within the same layer. The input and output vectors of DNN are represented by $x = \{x_1, x_2, \ldots, x_k\}^T$ and $Z = \{Z_1, Z_2, \ldots, Z_k\}^T$. The input vector $x$ consists of channel gains and transmission powers, whereas the output vector $Z$ consists of the sum rate for each cluster $j$. The number of nodes consisting of input, hidden, and output layers is represented by $I$, $H$, and $V$. The output layer predicts the sum rate information for the possible number of partial partitions for user grouping in each cluster $j$. The weights $W_1$ connect from the input layer to the hidden layer, and weights $W_{H+1}$ connect from the last hidden layer to the output layer.

In the $K$-user NOMA system, there are $2K$ input nodes and $K$ output nodes. The weight matrix is represented by $W^p$, which is $N_p x N_{p-1}$ that connects the $p-1$ layer to the $p$ layer. The activation function $f^p$ and the bias vector $b^p = [b_1^p, b_2^p, \ldots, b_k^p]^T$ for the $p$th layer respectively. The output vector $y^p$ of the hidden layer $a^p = [a_1^p, a_2^p, \ldots, a_k^p]^T$ for $1 \leq p \leq H$ can be expressed

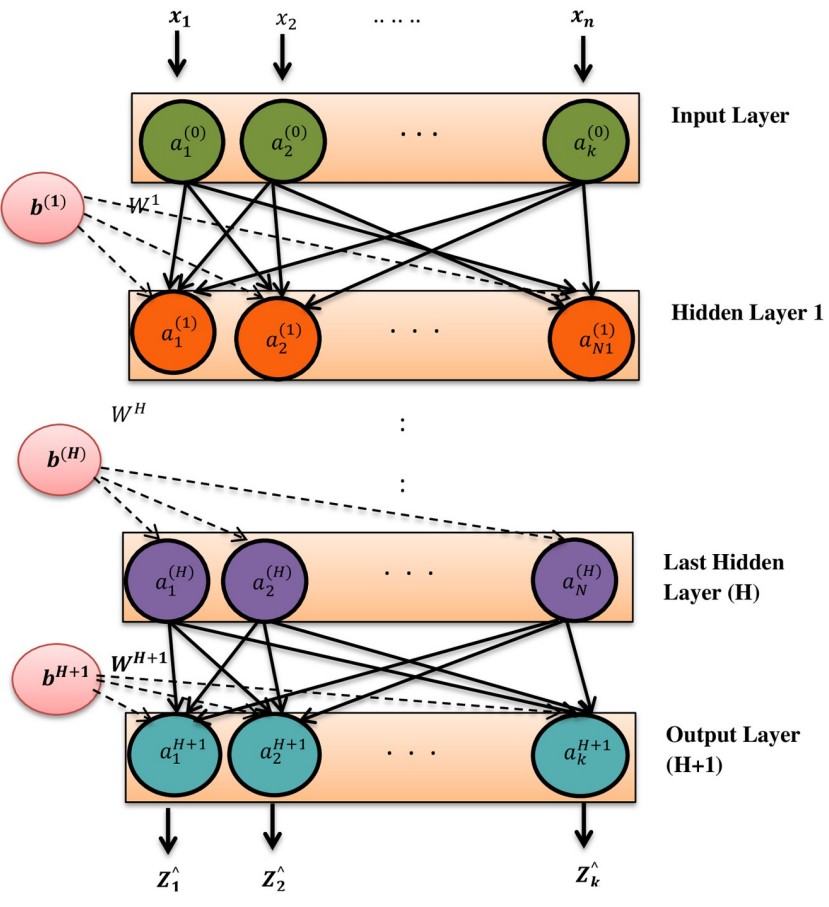

**Fig 3. DNN acrhitecture.**

as

$$a^p = f^p(y^p) \tag{23}$$

$$y^p = W^p a^{(p-1)} + b^p \tag{24}$$

In the proposed DNN method, the mean squared error (MSE) is chosen as the cost function, and it can be expressed as the final layer of the DNN is measured by using the aggregating weighted sum of $a^H$ and the output layer bias $b^H$, which can be expressed as

$$Z^* = W^{(H+1)} a^H + b^{(H+1)} \tag{25}$$

In the training phase, the process of finding the optimal user clustering is formulated to minimize the cost function. The proposed DNN-UC is trained using stochastic gradient descent, which iteratively updates the weights and biases via back-propagation as follows:

$$\alpha_{p+1} = \alpha_p - \varphi \nabla_\alpha H_\alpha \tag{26}$$

where $\alpha_p$ represents the parameter to be optimized, the subscript $p$ indicates the iteration number, $\varphi$ represents the scalar-valued step size, $\nabla_\alpha$ signifies the derivative with respect to $\alpha$, and $J_\alpha$ is the cost function. The learning process in the training phase of DNN is to find the optimal value of the sum rate by using a gradient descent method to update the weights and bias iteratively during back-propagation. In DNN-based P-BFS, the mean squared error. (MSE) is chosen as the cost function, and it can be expressed as

$$MSE = \frac{1}{D} \sum_{k=1}^{D} (Z_k - Z_k^*)^2 \tag{27}$$

Where $D$ is the total number of samples, $Z_k$ denotes the actual sum rate, and $Z_k^*$ represents the predicted sum rate of the $k^{th}$ user, which is generated by the partial BF-S method of each cluster $j$. The model's accuracy is measured by the ratio of corrected predicted sample to the total number of samples. If the MSE is large, the training data must be repeated more times until the MSE is as low as it should be. At the end of the network's training and validation processes, the model can be evaluated using the testing dataset.

## 4 Simulation results

In this section, the performance of IoT users in PD-NOMA is evaluated using the proposed resource allocation Algorithms 1 and 2.

The proposed user clustering results are obtained by considering the simulation parameters as shown in Table 3. The sum rate of $n$ = 10 users for possible partial partitions using the proposed P-BFS Algorithm 1 is shown in Fig 4. Using Algorithm 1, the number of partial partitions created for $n$ = 10 is 945 when considering $\psi$ = 10 for 2-user grouping, and the corresponding sum rate is shown in Fig 4. The cost matrix for the optimal sum rate value of $n$ IoT users inside cluster $j$ is computed using Algorithm 1. Fig 4 shows that partition number 494 gives a maximum sum rate compared to the others, so this partition is selected, and the others are neglected. Similarly, the dataset for $n > 10$ is generated by applying Algorithm 1 to find the optimal cluster and partition number that gives the maximum sum rate. The resultant user clustering cost matrix is obtained on the basis of the maximum sum rate of the IoT users inside the network.

In the proposed P-BFS method, the resultant cost matrix for user clustering is obtained by consuming a large amount of computation to search for the desired group of users in each

**Table 3. Simulation parameters.**

| Parameter | Value |
|---|---|
| Number of IoT users ($n$) | 10,20,30,40,50 |
| Number of users inside each group of cluster $j$ | 2,3,4 |
| Total System Bandwidth ($B$) | 5 MHz |
| Maximum BS transmission power downlink ($P_{max}$) | 43 dBm |
| Circuit Power ($P_c$) | 1 Watt |
| Radius of cell ($R$) | 500 m |
| Noise power spectral density ($N_o$) | -147 dBm/Hz |
| Minimum Rate ($R_{min}$) | 1 Mbps |
| Minimum power for SIC receiver ($\epsilon$) | 10 dBm |
| Path loss exponent (b) | 2.7 |

cluster $j$. After optimal user clustering, which is based on the proposed P-BFS method, our next step is to reduce the further complexity by applying DNN. We feed the UC dataset, which is obtained using Algorithm 1, to the DNN model to train the network for each number of cluster $j$. All IoT users are arranged in groups of 2 users in clusters $j$ on the basis of the maximum sum rate. The DNN extracts the features from a clustering data set and trains the network according to the simulation parameters, which are mentioned in Table 4. After training the network, it produces the output and verifies the predicted result with actual value using Eq 27. We distribute our UC data into 60% training, 20% validation, and 20% testing.

DNN is used to train the network and compute the sum rate of the system more efficiently in terms of time complexity. The purpose of applying the DNN-based proposed P-BFS is to reduce the further complexity of the P-BFS method by training the data set generated using

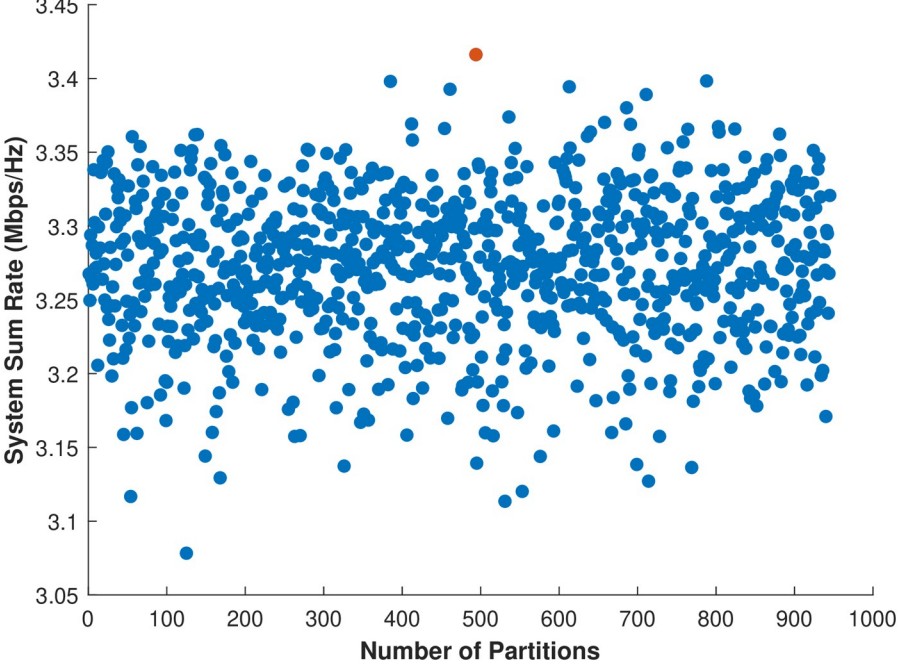

**Fig 4. Sum rate for partial partitions $n$ = 10.**

**Table 4. Simulation parameters for DNN.**

| Parameter | Value |
|---|---|
| Number of hidden Layers | 4 |
| Number of IoT users (n) | 10,20,30,40,50 |
| Size of samples | 945, 1890, 2835, 3780, 4725 |
| Size of training, validation and testing data | 70%, 15%, 15% |
| Learning rate | 0.001 |
| Number of input layer nodes | 20,40,60,80,1000 |
| Number of hidden layer nodes | 20,40,60,80,100 |
| Number of output layer nodes | 10,20,30,40,50 |
| Number of Epochs | 50,100,150,200 |
| Batch size | 10,20,30 |
| Activation function | ReLu |

Algorithm 1. The DNN model is trained according to the simulation parameters as shown in Table 4. We plot the actual and predicted sum rate values for $n = 10$ IoT users in cluster $j$ for selected samples from training, validation, and testing windows as shown in Fig 5. The addition of hidden layers greatly impacts the optimization of hyper-parameters, due to which DNN develops a more refined understanding for characterizing the nonlinearity of cluster formation, channel gains, and power. The results show in Fig 5 that the model efficiently trains the network to compute the predicted sum rate with the actual sum rate in each number of partitions for a specific cluster. The DNN-based approach optimally forms clusters, reducing the complexity of the proposed P-BFS method, with a sum rate closely approaching that of the actual proposed P-BFS user clustering scheme.

After the DNN-based proposed P-BFS user clustering, the next step is to find the optimal power of two users in each group of clusters $j$. KKT conditions are applied using Algorithm 2

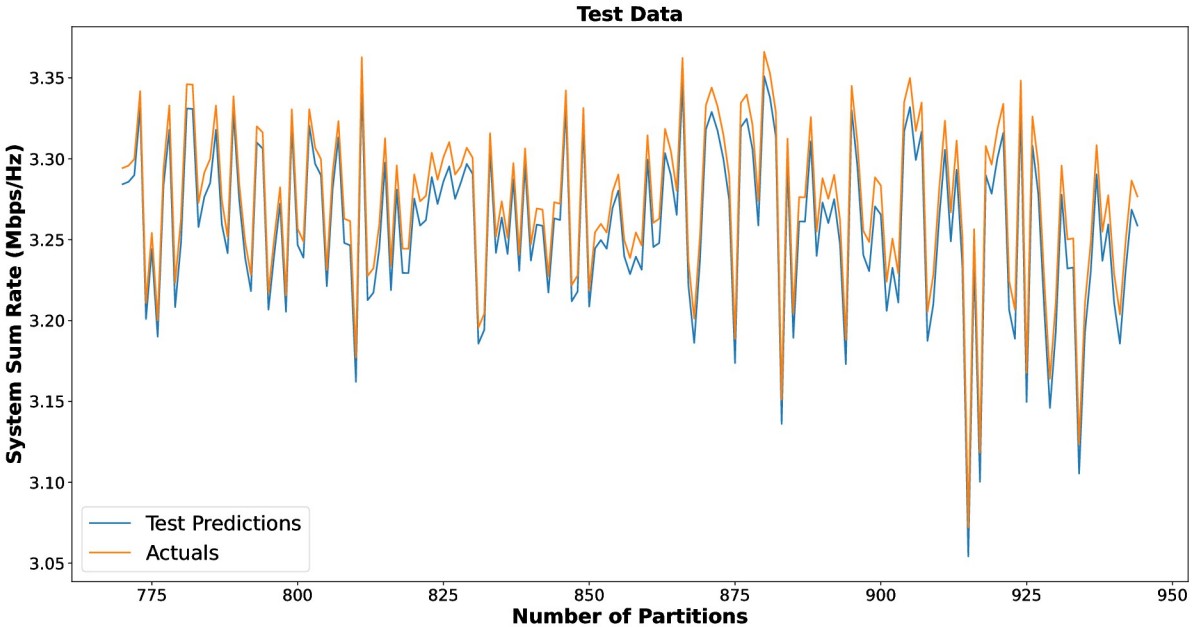

**Fig 5. DNN based testing result for $n = 10$.**

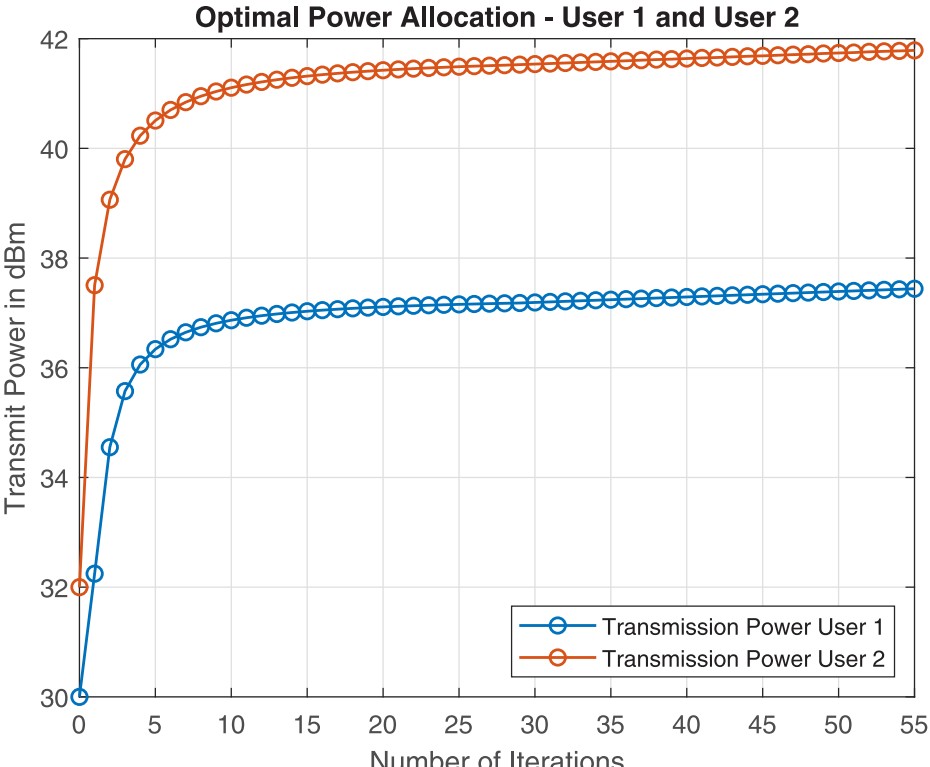

**Fig 6. Optimal power allocation in each cluster.**

to compute the optimal power value for the group of two users in each cluster of the proposed P-BFS. The simulation parameters are chosen from Table 3. The results in Fig 6 show that the optimal transmission power of user 1 is 37 dBm and the optimal power of user 2 is 42 dBm by employing Algorithm 2. Similarly, the optimal powers for the other users in each cluster are calculated. Fig 6 demonstrates that the power allocation is optimized significantly.

Fig 7 shows the sum rate of $n$ = 10 IoT users based on the optimal transmission power using KKT from 30 dBm to 42 dBm. The system sum rate of the proposed P-BFS for 2-user NOMA performs relatively better compared to 3-user and 4-user NOMA. The base station's transmission power varies from 1 watt to 16 watt, while the circuit power remains fixed at 1 watt. The system's energy efficiency performance, represented by the ratio of base station transmits power to circuit power, is significantly improved in the P-BFS 2-user NOMA scheme using Eq 5 compared to the other 3-user and 4-user grouping schemes.

Fig 8 shows the system's total data rate performance with varying numbers of IoT users. The range of these users is from 10 to 60 per base station. We observed that when the number of IoT users increases, the sum rate gradually improves in PD-NOMA. The plot in Fig 8 represents the sum rate for considering two cases of power transmission. In the first case, the transmission power is fixed at 30 dBm. In the second case, we consider the optimal transmission power by applying KKT conditions as mentioned in Algorithm 2, which provides an optimal value of transmission power of 42 dBm. The results in Fig 8 indicate that the P-BFS 2 user KKT performs well as compared to the other techniques.

In Fig 9, it is observed that as the number of IoT users ($n$) increases and the throughput performance of the DNN-based P-BFS scheme is relatively close to that of the proposed P-BFS

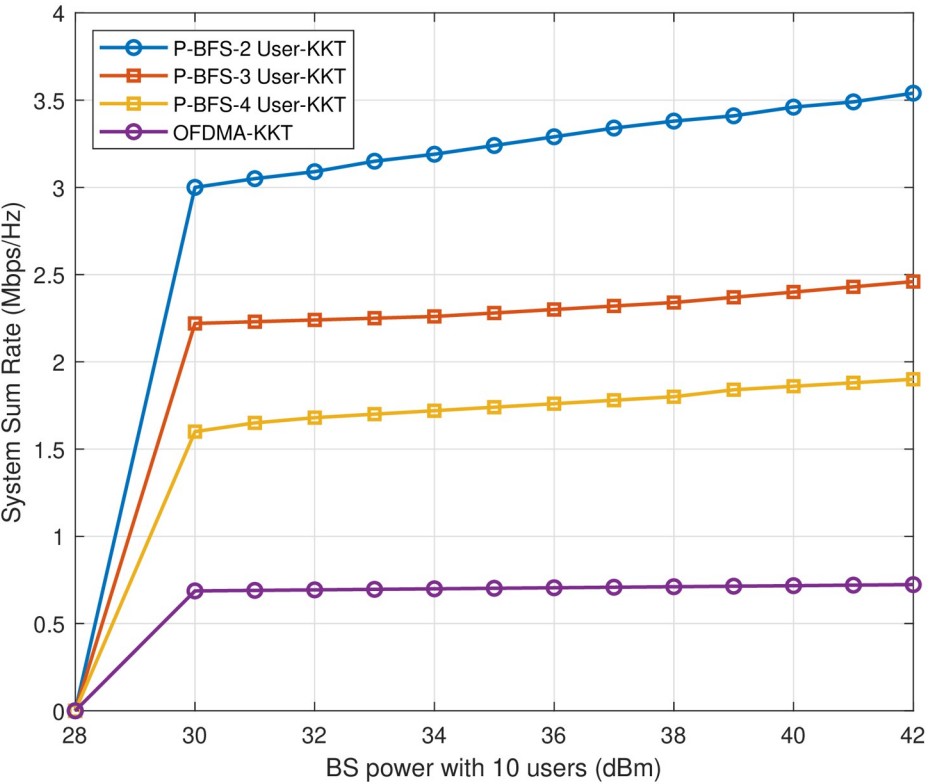

**Fig 7. Sum rate for optimal power $n = 10$.**

method for 2-user NOMA. There is a significant improvement in the sum rate after optimizing the transmission power of the BS using KKT, as shown in Fig 9. The optimal power allocation using KKT demonstrates a substantial increase in the sum rate with the growing ($n$) number of IoT users.

The proposed P-BFS-based user clustering is compared with an exhaustive BFS user clustering scheme in terms of system sum rate. The traditional exhaustive BFS is considered as a benchmark for user clustering data set in [7, 8]. The proposed P-BFS-based scheme consumes low computational cost in terms of complexity, and the sum rate performance of P-BFS is very close to the traditional exhaustive BFS method, as shown in Fig 10. We also compare the proposed P-BFS user clustering scheme with other heuristic channel-based user clustering (UC) proposed in [9, 10]. In user clustering, pairs of users are selected based on the highest difference in channel gains between two users, starting with the highest channel gain of the first user paired with the lowest channel gain of the last user, and so on. The same simulation and system parameters are chosen from Table 3 for user clustering comparison. Our approach outperforms other user clustering techniques as shown in Fig 10, offering substantial improvements in user clustering and resource assignment in NOMA-assisted IoT networks.

## 5 Conclusion

The efficient resource allocation is proposed in the PD-NOMA downlink scenario for IoT users in a network. We proposed a user clustering approach, which is based on P-BFS, to reduce the complexity comparatively in an exhaustive search technique. The cluster of

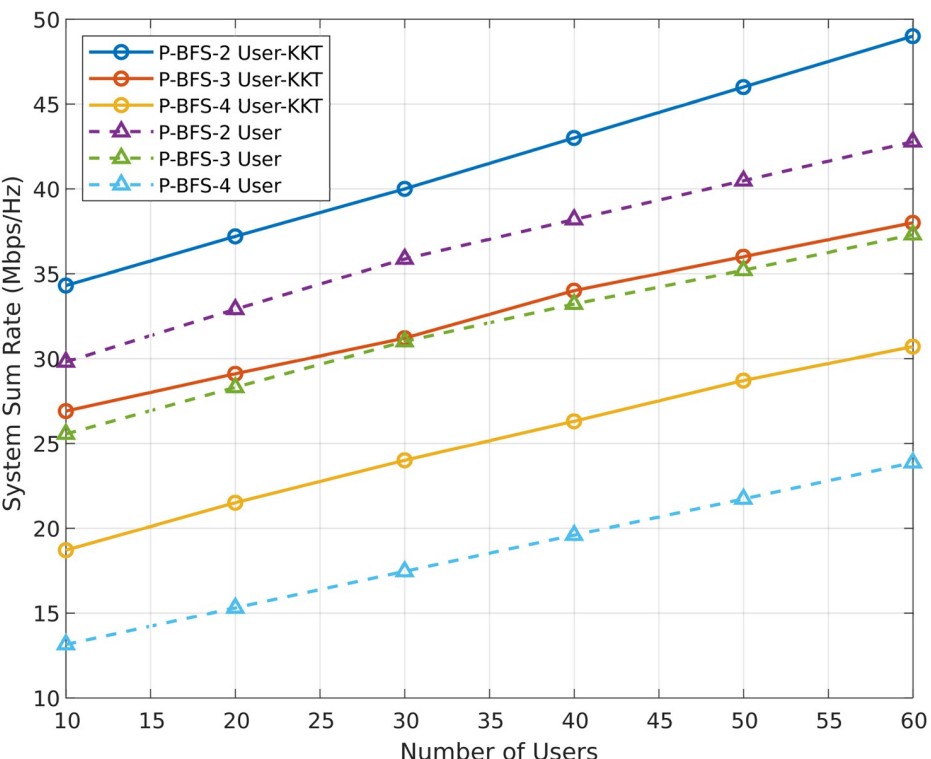

**Fig 8. Sum rate for number of users.**

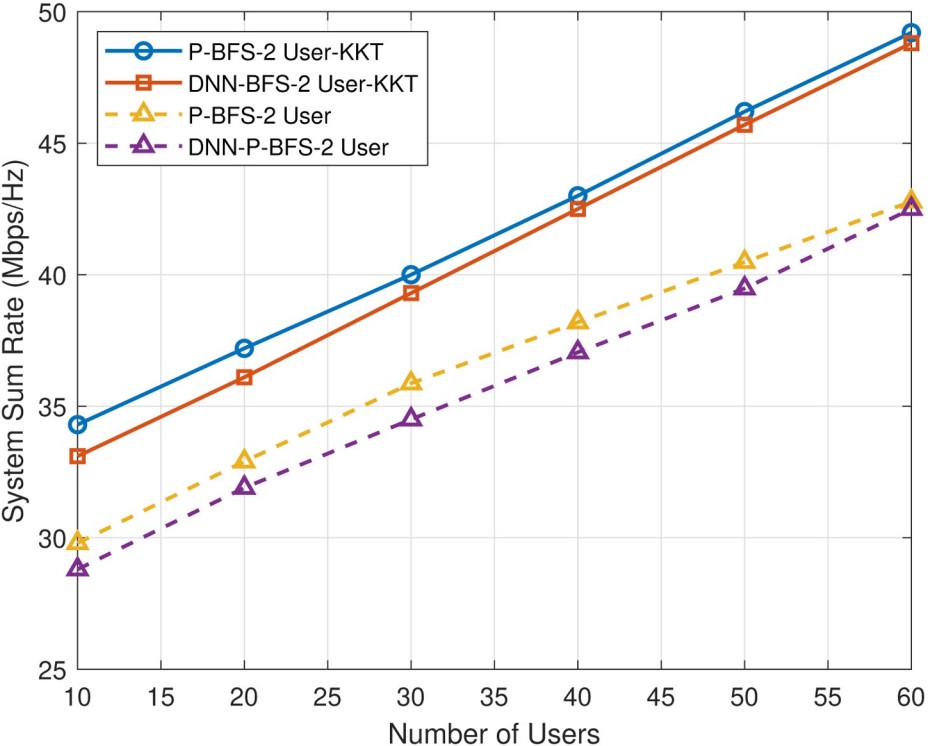

**Fig 9. Sum rate analysis of DNN based P-BFS NOMA for different number of users.**

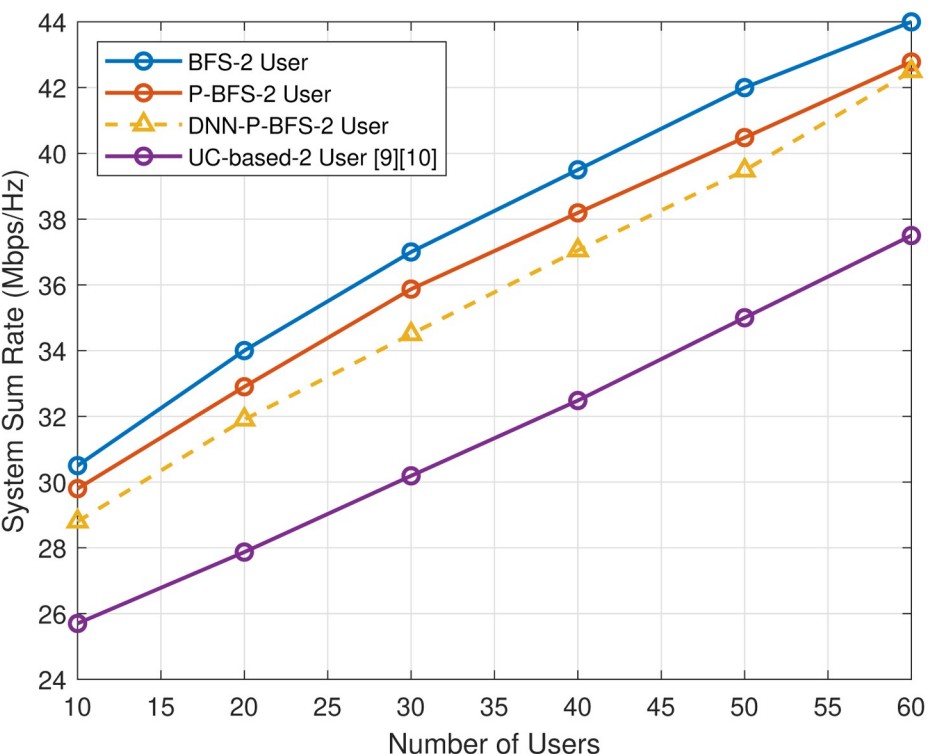

**Fig 10. User clustering techniques sum rate comparison.**

different groups of users, i.e., 2, 3, and 4, is selected on the basis of the maximum sum rate calculation in each possible partition using the proposed UC scheme. After clustering, the Lagrangian multiplier approach with Karush-Kuhn-Tucker optimal requirements is used to compute each user's optimal power allocation in each cluster. The power allocation is optimal, considering the QoS requirements of IoT users served by the BS. The proposed UC data set is fed to the deep neural network, which is used to train the network to make the clusters more efficient and reduce the further complexity of the proposed P-BFS method. The efficiency of DNN depends on the selection of the parameters: the number of hidden layers, epochs, and users inside each cluster. We observed from the results that the P-BFS prediction of user clustering is very close to the actual P-BFS. The simulation results show that the proposed P-BFS-based user clustering and optimal power calculation based on KKT provide the maximum sum rate to meet the quality of service requirement of IoT users in downlink PD-NOMA.

## Author Contributions

**Conceptualization:** Syed Muhammad Hamedoon, Jawwad Nasar Chattha, Muhammad Bilal.

**Data curation:** Syed Muhammad Hamedoon, Jawwad Nasar Chattha, Muhammad Bilal.

**Formal analysis:** Syed Muhammad Hamedoon, Jawwad Nasar Chattha, Muhammad Bilal.

**Funding acquisition:** Jawwad Nasar Chattha, Muhammad Bilal.

**Investigation:** Syed Muhammad Hamedoon, Jawwad Nasar Chattha, Muhammad Bilal.

**Methodology:** Syed Muhammad Hamedoon.

**Project administration:** Jawwad Nasar Chattha, Muhammad Bilal.

**Resources:** Syed Muhammad Hamedoon, Jawwad Nasar Chattha, Muhammad Bilal.

**Software:** Syed Muhammad Hamedoon.

**Supervision:** Jawwad Nasar Chattha, Muhammad Bilal.

**Validation:** Syed Muhammad Hamedoon, Jawwad Nasar Chattha.

**Visualization:** Syed Muhammad Hamedoon, Jawwad Nasar Chattha, Muhammad Bilal.

**Writing – original draft:** Syed Muhammad Hamedoon.

**Writing – review & editing:** Syed Muhammad Hamedoon, Jawwad Nasar Chattha, Muhammad Bilal.

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
