## [Decision Letter · Decision Letter 0]

25 Jul 2024

PONE-D-24-19575A Novel User Clustering and Efficient Resource Allocation in Non-Orthogonal Mutliple Access for IoT NetworksPLOS ONE

Dear Dr. Hamedoon,

Thank you for submitting your manuscript to PLOS ONE. After careful consideration, we feel that it has merit but does not fully meet PLOS ONE’s publication criteria as it currently stands. Therefore, we invite you to submit a revised version of the manuscript that addresses the points raised during the review process.

We look forward to receiving your revised manuscript.

Kind regards,

Mohammed Balfaqih

Academic Editor

PLOS ONE

 [This research work was funded by Institutional Fund Projects under grant no.

(IFPIP: 1822-135-1443). The authors gratefully acknowledge technical and financial support provided by the Ministry of Education and King Abdulaziz University, DSR, Jeddah, Saudi Arabia].  

[This research work was funded by Institutional Fund Projects under grant no.

(IFPIP: 1822-135-1443). The authors gratefully acknowledge technical and financial support provided by the Ministry of Education and King Abdulaziz University, DSR, Jeddah, Saudi Arabia]

 [This research work was funded by Institutional Fund Projects under grant no.

(IFPIP: 1822-135-1443). The authors gratefully acknowledge technical and financial support provided by the Ministry of Education and King Abdulaziz University, DSR, Jeddah, Saudi Arabia]

5. We note that your Data Availability Statement is currently as follows: [All relevant data are within the manuscript and its Supporting Information files.]

6. Please remove your figures from within your manuscript file, leaving only the individual TIFF/EPS image files, uploaded separately. These will be automatically included in the reviewers’ PDF.

Reviewers' comments:

Reviewer's Responses to Questions

**Comments to the Author**

1. Is the manuscript technically sound, and do the data support the conclusions?

Reviewer #1: Yes

Reviewer #2: Yes

2. Has the statistical analysis been performed appropriately and rigorously? 

Reviewer #1: Yes

Reviewer #2: Yes

3. Have the authors made all data underlying the findings in their manuscript fully available?

Reviewer #1: Yes

Reviewer #2: Yes

4. Is the manuscript presented in an intelligible fashion and written in standard English?

Reviewer #1: Yes

Reviewer #2: Yes

5. Review Comments to the Author

Reviewer #1: The authors presented an iterative optimization process, starting with user clustering and then moving to power allocation. The proposed user clustering technique used partial brute force search (P-BFS), whole power allocation was performed using the Lagrangian multiplier method with Karush-Kuhn-Tucker (KKT). The article is well written; however, the following issues must be addressed:

1-Huge use of constructions in first person. Examples: WE proposed…, WE calculate…, OUR work (abstract and introduction). Rewrite.

2- The references are extensive and relevant, but there is no clear explanation of how the literature review was conducted. A brief section detailing the bibliographic research methodology and the scientific bases used would enhance the transparency and credibility of the literature review process. References such as "An efficient method for resource allocation and user pairing in downlink non-orthogonal multiple access system" and " Filter orthogonal frequency‐division multiplexing scheme based on polar code in underwater acoustic communication with non‐Gaussian distribution noise" could be added.

3-It is not enough to explore the existing works without highlighting their limitations.

4-It is recommended that all symbols used in the article be compiled into a table for easy reference.

5-The proposed solution has not been compared with existing solutions to demonstrate its superiority.

Reviewer #2: The authors introduced an iterative optimization process, starting with user clustering followed by power allocation. Although the article is well-written, the following issues need to be addressed:

1.The assumptions made in the modeling, such as ideal propagation conditions, might not accurately reflect real-world scenarios. A discussion on how these assumptions impact the generalizability of the findings would be beneficial.

2.The paper does not provide a detailed comparison of the proposed methods with existing user clustering and resource allocation techniques, making it difficult to evaluate the relative superiority or efficiency of the new approach.

3.It is advised to compile all symbols used throughout the article into a table to facilitate easy reference.

6. PLOS authors have the option to publish the peer review history of their article (what does this mean?). If published, this will include your full peer review and any attached files.

Reviewer #1: No

Reviewer #2: No

---

## [Author Response · Author response to Decision Letter 0]

9 Aug 2024

Reviewer No. 1 : Thank you for reviewing the manuscript and for providing constructive feedback. We appreciate the time and effort you have dedicated to reviewing our work. 

Author Response sheet has attached along with the manuscript. All the track changes are in blue color in revised manuscript. Kindly see the attached author response document.

Reviewer No. 2 : Thank you for reviewing the manuscript and for providing constructive feedback. We appreciate the time and effort you have dedicated to reviewing our work. 

Author Response sheet has attached along with the manuscript. All the track changes are in blue color in revised manuscript. Kindly see the attached author response document.

---

## [Decision Letter · Decision Letter 1]

19 Aug 2024

A Novel User Clustering and Efficient Resource Allocation in Non-Orthogonal Mutliple Access for IoT Networks

PONE-D-24-19575R1

Dear Dr. Hamedoon,

We’re pleased to inform you that your manuscript has been judged scientifically suitable for publication and will be formally accepted for publication once it meets all outstanding technical requirements.

Kind regards,

Mohammed Balfaqih

Academic Editor

PLOS ONE

Additional Editor Comments (optional):

Reviewers' comments:

Reviewer's Responses to Questions

**Comments to the Author**

1. If the authors have adequately addressed your comments raised in a previous round of review and you feel that this manuscript is now acceptable for publication, you may indicate that here to bypass the “Comments to the Author” section, enter your conflict of interest statement in the “Confidential to Editor” section, and submit your "Accept" recommendation.

Reviewer #1: All comments have been addressed

Reviewer #2: All comments have been addressed

2. Is the manuscript technically sound, and do the data support the conclusions?

Reviewer #1: Yes

Reviewer #2: (No Response)

3. Has the statistical analysis been performed appropriately and rigorously? 

Reviewer #1: Yes

Reviewer #2: (No Response)

4. Have the authors made all data underlying the findings in their manuscript fully available?

Reviewer #1: Yes

Reviewer #2: (No Response)

5. Is the manuscript presented in an intelligible fashion and written in standard English?

Reviewer #1: Yes

Reviewer #2: (No Response)

6. Review Comments to the Author

Reviewer #1: All comments have been addressed. The authors have well restructured the paper and done good work. All points have been updated. The paper is now acceptable for publication.

Reviewer #2: (No Response)

7. PLOS authors have the option to publish the peer review history of their article (what does this mean?). If published, this will include your full peer review and any attached files.

Reviewer #1: No

Reviewer #2: No

---

## [Editor Report · Acceptance letter]

29 Aug 2024

PONE-D-24-19575R1 

PLOS ONE

Dear Dr. Hamedoon, 

I'm pleased to inform you that your manuscript has been deemed suitable for publication in PLOS ONE. Congratulations! Your manuscript is now being handed over to our production team.

Kind regards, 

on behalf of

Dr. Mohammed Balfaqih 

Academic Editor

PLOS ONE